# Robustify the Latent Space: Offline Distributionally Robust Reinforcement Learning with Linear Function Approximation

## Abstract

Among the reasons hindering the applications of reinforcement learning (RL) to real-world problems, two factors are critical: limited data and the mismatch between the test environment (real environment in which the policy is deployed) and the training environment (e.g., a simulator). This paper simultaneously addresses these issues with *offline distributionally robust RL*, where a distributionally robust policy is learned using historical data from the source environment by optimizing against a worst-case perturbation thereof. In particular, we move beyond tabular settings and design a novel linear function approximation framework that robustifies the latent space. Our framework is instantiated into two settings, one where the dataset is well-explored and the other where the dataset has weaker data coverage. In addition, we introduce a value shift algorithmic technique specifically designed to suit the distributionally robust nature, which contributes to our improved theoretical results and empirical performance. Sample complexities $\tilde{O}(d^{1/2}/N^{1/2})$ and $\tilde{O}(d^{3/2}/N^{1/2})$ are established respectively as the first non-asymptotic results in these settings, where $d$ denotes the dimension in the linear function space and $N$ represents the number of trajectories in the dataset. Diverse experiments are conducted to demonstrate our theoretical findings, showing the superiority of our algorithms against the non-robust one.

## 1 Introduction

Unlike data-driven methods in supervised learning, reinforcement learning (RL) algorithms require active interaction with the environment to learn a near-optimal policy, often involving online trial-and-error. However, this approach can be impractical in real-world scenarios with limited or prohibited data collection. To address the limitation of online RL, offline reinforcement learning (offline RL or batch RL) (Lange et al., 2012; Levine et al., 2020), focuses on policy learning with only access to some logged datasets and expert demonstrations. Due to its non-dependence on further interaction with the environment, offline RL is increasingly appealing for various applications, including autonomous driving (Yu et al., 2018; Yurtsever et al., 2020; Shi et al., 2021), healthcare (Gottesman et al., 2019; Yu et al., 2021; Tang & Wiens, 2021) and robotics (Siegel et al., 2020; Zhou et al., 2021a; Rafailov et al., 2021).

Despite the developments in the rich literature (Yu et al., 2020; Kumar et al., 2020; Yang et al., 2021b; An et al., 2021; Cheng et al., 2022), offline RL has an implicit but questionable assumption: the test environment is the same as the training one. This assumption can result in inadequate performance of offline RL in uncertain environments since the optimal policy of a Markov decision process (MDP) may be sensitive to the transition probabilities (Mannor et al., 2004; El Ghaoui & Nilim, 2005; Simester et al., 2006). Including financial trading and robotics, many domains may prefer a robust policy that remains effective in shifting distributions from the one in the training environment. Thus, robust MDPs have been proposed to address this issue (Satia & Lave Jr, 1973; Nilim & El Ghaoui, 2005; Iyengar, 2005; Wiesemann et al., 2013; Lim et al., 2013; Ho et al., 2021; Goyal & Grand-Clement, 2022). Recent studies (Zhou et al., 2021b; Yang et al., 2021a; Shi & Chi, 2022; Panaganti et al., 2022a) demonstrate the potential of robust RL in the offline setting.

In this paper, we aim to theoretically understand linear function approximation as an important component in offline distributionally (DR) robust RL. Linear function approximation (Bertsekas & Tsitsiklis, 1995; Schweitzer & Seidmann, 1985), which uses a linear combination of features to approximate the value function, is one of the most widely used and studied solutions in high-dimensional problems and serves as a cornerstone in the path toward large-scale real-world problems.

Developing a DRRL algorithm with linear function approximation is challenging. In contrast to non-robust RL where transition kernel is fixed, DRRL assumes that the transition kernel of the MDP belongs to an ambiguity set, which significantly impacts the computational feasibility of the robust value function and policy performance. A common approach in this area is to construct the ambiguity set for each state-action pair and then project the robust value function onto a lower-dimensional subspace using linear function approximation, called Robustify-then-Approximate (RTA) approach in this paper. Although RTA style algorithms (Wiesemann et al., 2013; Goyal & Grand-Clement, 2022; Panaganti et al., 2022a) have proven that their robust projected value iteration can converge to a fixed point, as pointed out by our motivating example in Section 3, linear projection may conflict with the non-linearity of the robust Bellman operator, which may consequently lead to suboptimal decisions. Furthermore, none of these algorithms have been shown to be sample efficient under weak data conditions, which is essential in real-world applications. Recently, a study by Goyal & Grand-Clement (2022) proposes constructing the ambiguity set in the latent space (called Robustify-the-Latent Space (RLS)), which is more compatible with the linear approximator. However, they assume access to the true transition kernel, whereas in practice, we can only access data sampled from some training environment. Therefore, developing a data-driven DRRL algorithm that directly robustifies the latent space is yet to be explored.

In this paper, we mainly address the question below building on insights from high-dimensional statistics (Wainwright, 2019) to provide informative insights into the impacts of salient problem parameters on the sample complexity, especially for applications with large state-action spaces:

*Is it possible to design a sample-efficient algorithm using linear function approximation for offline DRRL by robustifying the latent space, even with weaker data coverage conditions?*

We give a positive response to this question. Specifically, our contributions are fourfold:

1. We point out potential conflicts between linear function approximation and robustness gain in the Robustify-then-approximate (RTA) approach by constructing a motivating example. Then we instantiate the idea of robustifying the latent space (RLS) into a sample-efficient **D**istributionally **R**obust **V**alue **I**teration with **L**inear function approximation (DRVI-L) algorithm for well-explored datasets.

2. We prove a state-action space independent sample complexity guarantee for our DRVI-L algorithm with a novel value shift technique to alleviate the magnification of the estimation error in the DR optimization nature. This result can almost recover to the same dependence on $d$ and $N$ of that from the non-robust counterpart (Yin et al., 2022), which has never been achieved by previous literature.

3. We extend our algorithm by designing the **P**essimistic **D**istributionally **R**obust **V**alue **I**teration with **L**inear function approximation (PDRVI-L) algorithm, a pessimistic variant with our DRVI-L, and prove the first sample-efficient bound beyond the well-explored condition. Such an extension is the fruit of our RLS idea with delicate non-asymptotic analysis.

4. We establish theoretical guarantees for our two algorithms even when the MDP transition is not perfectly linear, and conduct experiments to demonstrate the balance achieved by our linear function approximation algorithm between optimality and computational efficiency.

## 2 Preliminary

### 2.1 MDP structure and Notations

Consider an episode MDP $(\mathcal{S}, \mathcal{A}, H, \mu, P, r)$ where $\mathcal{S}$ and $\mathcal{A}$ are finite state and action spaces with cardinalities $S$ and $A$. $P = \{P_h\}_{h=1}^{H}$ are state transition probability measures and $r = \{r_h\}_{h=1}^{H}$ are the reward functions, respectively. We assume that $r$ is deterministic and bounded in $[0, 1]$. A (Markovian) policy $\pi = \{\pi_h\}_{h=1}^{H}$ maps, for each period state-action pair $(s, a)$ to a probability

distribution over the set of actions $\mathcal{A}$ and induce a random trajectory $s_1, a_1, r_1, \cdots, s_H, a_H, r_H, s_{H+1}$ with $s_1 \sim \mu$, $a_h \sim \pi(\cdot|s_h)$and $s_{h+1} \sim P_h(\cdot|s_h, a_h)$ for $h \in [H]$ for some initial state distribution $\mu$. For any policy $\pi$ and any stage $h \in [H]$, the value function $V_h^\pi : \mathcal{S} \to \mathbb{R}$, the action-value function $Q : \mathcal{S} \times \mathcal{A} \to \mathbb{R}$, the expected return $R(\pi, P)$ are defined as $V_h^\pi(s) := \mathbb{E}_P^\pi[\sum_{h=1}^H r_h(s_h, a_h)|s_h = s]$, $Q_h^\pi(s, a) := \mathbb{E}_P^\pi[\sum_{h=1}^H r_h(s_h, a_h)|s_h = s, a_h = a]$, and $R(\pi, P) := \mathbb{E}_{s \sim \mu}[V_1^\pi(s)]$. For any function $Q$ and any policy $\pi$, we denote $\langle Q(s, \cdot), \pi(\cdot|s) \rangle_{\mathcal{A}} = \sum_{a \in \mathcal{A}} Q(s, a)\pi(a|s)$. For two non-negative sequences $\{a_n\}$ and $\{b_n\}$, we denote $\{a_n\} = O(b_n)$ if $\limsup_{n \to \infty} a_n/b_n < \infty$. We also use $\tilde{O}(\cdot)$ to denote the respective meaning within multiplicative logarithmic factors in $N$, $d$, $H$ and $\delta$. We denote the Kullback-Leibler (KL) divergence between two discrete probability distributions $P$ and $Q$ over state space as $D_{\mathrm{KL}}(P\|Q) = \sum_{s \in \mathcal{S}} P(s) \log(\frac{P(s)}{Q(s)})$.

## 2.2 DISTRIBUTIONALLY ROBUST MDPs

Before we present the DRRL setting, we first introduce the Distributionally Robust MDPs (DRMDPs). DRMDPs assume that the probability $P$ is not exactly known but lies within a so-called ambiguity set $\mathcal{P}$ induced by a distribution distance measure, such as KL divergence. The return of any given policy is the worst-case return induced by the transition model over the ambiguity set. We define the DR value function, action-value function and expected return as $V_h^{\pi,\mathrm{rob}}(s) = \inf_{P \in \mathcal{P}} V_h^\pi(s)$, $Q_h^{\pi,\mathrm{rob}}(s, a) = \inf_{P \in \mathcal{P}} Q_h^\pi(s, a)$ and $R^{\mathrm{rob}}(\pi, \mathcal{P}) = \inf_{P \in \mathcal{P}} R(\pi, P)$), respectively. The optimal DR expected return is defined as $R^{\mathrm{rob}}(\pi^*, \mathcal{P}) := \sup_{\pi \in \Pi} R^{\mathrm{rob}}(\pi, \mathcal{P})$ over all Markovian policies. In the sequel, we omit the superscript "rob". In fact, by the work of Goyal & Grand-Clement (2022), we can restrict to the deterministic policy class to achieve the optimal DR expected return. The performance metric for any given policy $\pi$ is the so-called suboptimality, which is defined as

$$\mathrm{SubOpt}(\pi; \mathcal{P}) = R(\pi^*, \mathcal{P}) - R(\pi, \mathcal{P}).$$

As the transition models and the policies are sequences corresponding to all horizons in the episode MDP, following Iyengar (2005), we assume that $\mathcal{P}$ can be decomposed as the product of the ambiguity sets in each horizon, i.e., $\mathcal{P} = \prod_{h=1}^H \mathcal{P}_h$. For stage $h$, each transition model $P_h$, lies within the ambiguity set $\mathcal{P}_h$.

## 2.3 LINEAR MDP

Our main task in this paper is to compute the optimal policy using linear function approximation with the possible lowest suboptimality. We parameterize the Q-function, value function, and optimal policy for each horizon $h \in [H]$ using $\nu_h \in \mathbb{R}^d$, given the feature map $\phi : \mathcal{S} \times \mathcal{A} \to \mathbb{R}^d$, as follows:

$$Q_{\nu_h}(s, a) := \phi(s, a)^\top \nu_h, \quad V_{\nu_h}(s) := \max_{a \in \mathcal{A}} Q_{\nu_h}(s, a), \quad \pi_{\nu_h}(s) := \arg\max_{a \in \mathcal{A}} Q_{\nu_h}(s, a). \quad (1)$$

To study the linear function approximation, various assumptions on the MDP have been proposed in the literature (Jiang et al., 2017; Yang & Wang, 2019; Jin et al., 2020; Modi et al., 2020; Zanette et al., 2020; Wang et al., 2021). In particular, we consider the soft state aggregation of $d$ factors, i.e., the transition model in each stage $h$ can be represented using a known feature map $\phi : \mathcal{S} \times \mathcal{A} \to \mathbb{R}^d$ over the $d$ latent factor spaces defined by $\psi_h : \mathcal{S} \to \mathbb{R}^d$. Such a assumption has been widely adopted in the literature (Singh et al., 1994; Duan et al., 2019; Zhang & Wang, 2019; Zanette et al., 2021). We also assume the reward functions admit linear structure w.r.t. $\phi$, following the linear MDP protocol from Jin et al. (2021; 2020). Formally, we have the following definition for the soft state aggregation.

**Definition 2.1** (Soft State Aggregation MDP). *Consider an episode MDP instance $M = (\mathcal{S}, \mathcal{A}, H, P, r)$ and a feature map $\phi : \mathcal{S} \times \mathcal{A} \to \mathbb{R}^d$. We say the transition model $P$ admit a soft state aggregation w.r.t. $\phi$ (denoted as $P \in \mathrm{Span}(\phi)$) if for every $s \in \mathcal{S}, a \in \mathcal{A}, s' \in \mathcal{S}$ and every $h \in [H]$, we have*

$$P_h(s'|s, a) = \phi(s, a)^\top \psi_h(s'),$$

*for some factors $\psi_h : \mathcal{S} \to \mathbb{R}^d$. Moreover, $\psi$ satisfies,*

$$\int_s \psi_{h,i}(s)ds = 1, \forall i \in [d], h \in [H].$$

*We say the reward functions $r$ admit a linear representation w.r.t. $\phi$ (denoted as $r \in \mathrm{Span}(\phi)$) if for all $s \in \mathcal{S}$ and $a \in \mathcal{A}$ and $h \in [H]$, there exists $\theta_h \in \mathbb{R}^d$ satisfying $\|\theta_h\| \leq \sqrt{d}$ and $r_h(s, a) = \phi(s, a)^\top \theta_h$.*

## 3 MOTIVATING EXAMPLE

Computing an optimal robust policy for a general ambiguity set is proven to be strongly NP-Hard, as demonstrated by Wiesemann et al. (2013). To ensure computational tractability, Nilim & El Ghaoui (2005) and Iyengar (2005) introduce the $(s, a)$-rectangular ambiguity set. This set assumes that the perturbation of the transition probability for each $(s, a)$ pair is independent of others, denoted as $\mathcal{P}_h = \prod_{(s,a) \in \mathcal{S} \times \mathcal{A}} \mathcal{P}_h(s, a)$. However, this assumption can be computationally expensive when dealing with large state or action spaces since it requires solving a robust optimization problem for each $(s, a)$ pair. Moreover, it may exhibit over-conservatism, particularly when the transition probabilities possess inherent structure (Wiesemann et al., 2013; Goyal & Grand-Clement, 2022).

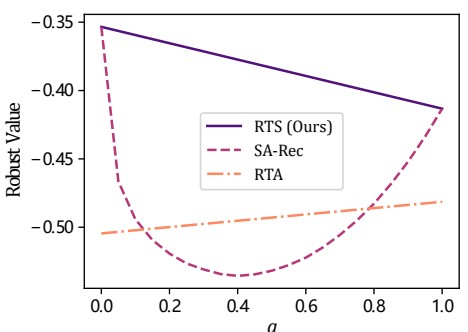

Figure 1: Motivating example. See Appendix B for the detailed experiment setup.

Before this work, the only attempts in linear function approximation for the robust RL are Tamar et al. (2014); Badrinath & Kalathil (2021). Both approaches follow the same idea: first obtaining robust values for each $(s, a)$-pair and then approximating robust values for the entire state-action space using a linear function. We refer to this as robustify-then-approximate (RTA). In contrast, our proposed algorithm directly robustifies the latent space (RTS). We illustrate the limitations of RTA using a continuous bandit case, which corresponds to the offline DRRL scenario with $H = 1$ and $S = 1$, where the action set is $[0, 1]$. When selecting action $a = 0$, the reward $r_0$ is drawn from a normal distribution with mean 1 and variance 1. If the action $a = 1$ is chosen, the reward $r_1$ follows a normal distribution with mean 0 and variance 0.5. When $a \in (0, 1)$, the reward distribution $r_a$ is a mixture of $r_0$ and $r_1$, with a probability of $(1 - a)$ and $a$, respectively.

Given the linear structure of the problem, it is desirable to use linear function approximation to maintain a low-dimensional representation. However, as shown in Figure 1, the projected robust values using Tamar et al. (2014); Badrinath & Kalathil (2021)'s methods are irrational due to the nonlinear nature of the $(s, a)$-uncertainty set. Specifically, the projected algorithm behaves more pessimistically than the $(s, a)$-rectangular method for actions close to 0 or 1. It even **fails to preserve the order relationship between the robust values of actions** 0 **and** 1, as the robust value of action 0 is higher than 1 in the $(s, a)$-rectangular but becomes lower after projection. This loss of order may lead to suboptimal decisions, while our proposed algorithm, based on the $d$-rectangular ambiguity set (defined in Section 4), recovers the robust values for action 0 and 1 while avoiding the over-conservatism for action in between.

## 4 LINEAR FUNCTION APPROXIMATION: ROBUSTIFY THE LATENT SPACE

In this section, we assume the MDP enjoys the soft state-aggregation structure and introduce the concrete approach to robustify the latent space, i.e., the so-called $d$-rectangular ambiguity set. We then propose our first algorithm, Distributional Robust Value Iteration with Linear function approximation (DRVI-L) with corresponding sample complexity results.

**Assumption 4.1** (State-Aggregation MDP). *The true transition models are soft state-aggregation w.r.t. $\phi$ and the reward functions admit linear representation w.r.t. $\phi$ (Definition 2.1).*

### 4.1 AMBIGUITY SET STRUCTURE: ROBUSTIFY THE LATENT SPACE

To robustify the latent factor space, we assume each factor lies in an ambiguity set, which is formally stated in the Assumption 4.2.

**Assumption 4.2** (*d*-rectangular). *For each $h \in [H]$, we assume the ambiguity set $\mathcal{P}_h$ with radius $\rho$ admits the following structure for some probability distance $D : \mathbb{R}^d \times \mathbb{R}^d \to \mathbb{R}_{\geq 0}$,*

$$\mathcal{P}_h(\rho) = \{(\sum_{i \in [d]} \phi_i(s,a)\psi'_{h,i}(s'))_{sas'} : \forall D(\psi'_{h,i}, \psi_{h,i}) \leq \rho\}.$$

Under Assumption 4.2, each factor $\psi_{h,i}$ is assumed to be independent, and thus can be chosen arbitrarily within the set $\mathcal{P}(\psi_h; \rho) := \{\psi'_{h,i} : D(\psi'_{h,i}, \psi_{h,i}) \leq \rho\}$ without affecting other factors. We formulate our offline DRRL problem with $d$-rectangular as:

$$R(\pi; \mathcal{P}) = \inf_{P \in \mathcal{P}(\rho) = \prod_{h=1}^{H} \mathcal{P}_h(\rho)} R(\pi, P). \tag{2}$$

To facilitate algorithm design and suboptimality analysis, we choose the KL divergence as the probability distance function $D$. The corresponding KL ambiguity set for a horizon $h$ and the $i$-th factor is denoted as $\mathcal{P}^{\mathrm{KL}}(\psi_{h,i}; \rho)$, while the ambiguity set for the horizon $h$ is denoted as $\mathcal{P}_h^{\mathrm{KL}}(\rho) := \prod_{i \in [d]} \mathcal{P}^{\mathrm{KL}}(\psi_{h,i}; \rho)$. To ensure computational tractability, we use Lemma 4.1, a strong duality result proven by Hu & Hong (2013), which allows us to solve the primal problem over the KL ambiguity set by solving the one-dimensional dual problem over the dual function $\sigma(Z, \beta)$.

**Lemma 4.1** (Hu & Hong (2013)). *Suppose $X \sim P$ has finite moment generating function in the neighborhood of zero. We denote $Z := \mathbb{E}_P[e^{-X/\beta}]$ and the dual function $\sigma(Z, \beta) := -\beta \log(Z) - \beta \cdot \rho$, then,*

$$\inf_{P' : D_{\mathrm{KL}}(P' \| P) \leq \rho} \mathbb{E}_{P'}[X] = \sup_{\beta \geq 0} \sigma(Z, \beta). \tag{3}$$

As $\rho \to 0$, the LHS of the dual equation degrades to the non-robust view, i.e., $\mathbb{E}_P[X]$, and the optimum $\beta^* = \arg\sup_{\beta \geq 0} \sigma(Z, \beta)$ approaches infinity.

Based on Lemma 4.1 and Assumption 4.1, we can derive the following DR Bellman operator:

$$(\mathbb{B}_h V)(s,a) = r_h(s,a) + \inf_{P_h \in \mathcal{P}_h^{\mathrm{KL}}(\rho)} \mathbb{E}_{s' \sim P_h(\cdot|s,a)}[V(s')] = \phi(s,a)^\top (\theta_h + w_h), \tag{4}$$

where $w_{h,i} = \sup_{\beta \geq 0} \sigma(\mu_{h,i}, \beta)$ and $\mu_{h,i} := \mathbb{E}_{\psi_{h,i}}[e^{-V(s')/\beta}] = \int_{s'} \psi_{h,i}(s')e^{-V(s')/\beta}ds'$.

The preceding result indicates that the DR Bellman operator using the $d$-rectangular ambiguity set can maintain $\phi$ representation. We formally state it in Lemma 4.2.

**Lemma 4.2.** *For any policy $\pi$ and any epoch $h \in [H]$, the DR Q-function is linear w.r.t. $\phi$. Moreover, $d(\mathbb{B}_h\mathcal{F}, \mathcal{F}) = 0$, where $d(\mathbb{B}_h\mathcal{F}, \mathcal{F}) = \sup_{g \in \mathcal{F}} \inf_{f \in \mathcal{F}} \|f - \mathbb{B}_h g\|$ is the Bellman error (Munos & Szepesvári, 2008) and $\mathcal{F} = \{\phi(\cdot, \cdot)^\top w : \forall w \in \mathbb{R}^d\}$ is a set containing all the possible function using $\phi$ as the feature map.*

Lemma 4.2 forms the basis for our function approximation algorithmic design. Unlike previous literature, such as Panaganti et al. (2022a), which assumes the completeness of the function class with respect to the $\phi$ representation without verification, our approach addresses the incompleteness issue observed in Section 3 by robustifying the feature space.

## 4.2 DISTRIBUTIONALLLY ROBUST VALUE ITERATION WITH VALUE SHIFT

The key challenge in offline (DR)RL problem is that the computation of the DRRL policy is restricted to only utilize a logged dataset, rather than having access to the exact transition probability or interaction with the environment. As a result of the lack of ongoing interaction with the environment, the performance of the offline RL algorithm is adversely affected by the insufficient coverage of the offline dataset. As a starting point, we consider a robust-variant of the uniform "well-exploration" condition, which is widely adopted in many offline RL works (Jin et al., 2021; Duan et al., 2019; Xie et al., 2021).

**Assumption 4.3** (Uniformly Well-explored Dataset). *Suppose $\mathcal{D}$ consists of $N$ trajectories $\{(s_h^\tau, a_h^\tau, r_h^\tau)\}_{\tau,h=1}^{N,H}$ independently and identically induced by a fixed behavior policy $\bar{\pi}$ in the linear MDP. Meanwhile, suppose there exists an absolute constant $\underline{c} > 0$ such that at each step $h \in [H]$ and any $P \in \mathcal{P}(\rho)$*

$$\lambda_{\min}(\Sigma_h^P) \geq \underline{c}, \quad \text{where } \Sigma_h^P = \mathbb{E}_{P,\bar{\pi}}[\phi(s_h, a_h)\phi(s_h, a_h)^\top].$$

Such an assumption requires the behavior policy to explore each feature dimension well, even in the worst-case transition model, which might need to explore some state-action pairs that the optimal policy has seldom visited. Similar assumption has appeared in (Shi & Chi, 2022) in the tabular setting.

To approximate the true Bellman operator, we construct the empirical version of Equation 4, particularly, to approximate $\mu_{h,i}$. Notably,

$$\mathbb{E}_{P_{s,a}}[e^{-V(s')/\beta}] = \int_{s'} e^{-V(s')/\beta} P(s'|s,a)ds' = \phi(s,a)^\top \mu_h,$$

where samples from $P_{s,a}$ can be obtained, motivating us to approximate $\mu_h$ by linear regression to obtain the estimator $\widehat{\mu}_h$. Note that $\mu_{h,i}$ and $\widehat{\mu}_{h,i}$ could be very close to zero, and further cause $\sigma(\widehat{\mu}_{h,i}, \beta)$ to approach infinity and damage the estimation. To address this issue, we propose a novel value shift technique by defining a new dual function $\tilde{\sigma}(Z, \beta)$ by changing $Z$ to $Z + 1$,

$$\tilde{\sigma}(Z, \beta) = -\beta \log(Z + 1) - \beta \cdot \rho. \tag{5}$$

This ensures that $\log(Z + 1)$ remains valid even $Z$ approaches zero. Accordingly, we adopt the shifted variant of the regression objective by subtracting 1 from the regression target, defined as

$$\tilde{\mathcal{E}}_h(\mu) = \sum_{\tau=1}^N ((e^{-V(s_{h+1}^\tau)/\beta} - 1) - \phi(s_h^\tau, a_h^\tau)^\top \mu)^2,$$

$$\widehat{\mu}_h = \arg\min_{\mu \in \mathbb{R}^d} \tilde{\mathcal{E}}_h(\mu) + \lambda \cdot \|\mu\|^2, \quad \widehat{w}_{h,i} = \sup_{\beta \geq 0} \tilde{\sigma}(\min\{(\widehat{\mu}_{h,i})_+, 1\}, \beta). \tag{6}$$

We define $\Lambda_h = \sum_{\tau=1}^N \phi(s_h^\tau, a_h^\tau) + \lambda \cdot I$. $\widehat{\theta}_h$ and $\widehat{w}_h$ have the closed form as

$$\widehat{\theta}_h = \Lambda_h^{-1}(\sum_{\tau=1}^N \phi(s_h^\tau, a_h^\tau) r_h^\tau), \quad \widehat{\mu}_h = \Lambda_h^{-1}(\sum_{\tau=1}^N \phi(s_h^\tau, a_h^\tau)(e^{-V(s_{h+1}^\tau/\beta)} - 1)).$$

Our value shifting technique ensures that $\widehat{w}_h$ maintains a valid value regardless of the estimator's quality, which is essential for achieving the desired suboptimality that can nearly recover to that of the non-robust setting. We summarize our algorithm as **D**istributional **R**obust **V**alue **I**teration with **L**inear function approximation (DRVI-L) in Algorithm 1.

Prior to presenting the suboptimality analysis for our Algorithm 1, we introduce Assumption 4.4, which assumes a known, common lower bound for the optimum of the KL optimization problem in Lemma 4.1. This assumption is also necessary in the tabular case (Zhou et al., 2021b; Panaganti & Kalathil, 2022).

**Assumption 4.4.** *For each $h \in [H]$ and each $i \in [d]$, we denote $\beta_{h,i}^* = \arg\sup_{\beta_{h,i} \geq 0} \sigma(\mu_{h,i}, \beta_{h,i})$. We assume there exists a known $\underline{\beta}$ s.t. $0 < \underline{\beta} \leq \min_{h \in [H], i \in [d]} \beta_{h,i}^*$.*

By Proposition 2 in Hu & Hong (2013), $\beta_{h,i}^* = 0$ when the worst case happens with sufficient large probability w.r.t. $\rho$. In practice, it is typical to employ a small value of $\rho$ to adapt to the problem without incurring over-conservatism (Ben-Tal & Nemirovski, 1998; 2000; Duchi & Namkoong, 2021). Thus $\beta_{h,i}^*$ would rarely be zero and enjoy a common non-zero lower bound.

**Theorem 4.1.** *We set $\lambda = 1$ in Algorithm 1. Under the Assumption 4.1, Assumption 4.3 and Assumption 4.4, when $N \geq 40/\underline{c} \cdot \log(4dH/\delta)$, we have the following holds with probability at least $1 - \delta$,*

$$\text{SubOpt}(\widehat{\pi}; \mathcal{P}) \leq c_1 \underline{\beta}(e^{H/\underline{\beta}} - 1)d^{1/2}\zeta_1^{1/2}H/N^{1/2} + c_2\underline{\beta}^{1/2}(e^{H/\underline{\beta}} - 1)\zeta_2^{1/2}H^{3/2}/N^{1/2}.$$

*Here $\zeta_1 = \log(2N + 16Nd^{3/2}H^2e^{H/\underline{\beta}})$, $\zeta_2 = \log(\frac{2dNH^3}{\delta\rho})$ and $c_1$ and $c_2$ are some absolute constants that only depend on $\underline{c}$.*

It is worth noting that the suboptimality of Algorithm 1 primarily depends on the dimension $d$ rather than the size of the state-action space. In contrast to tabular cases, such as Zhou et al. (2021b); Yang et al. (2021a), which focus on bounding the finite sample error for individual $(s, a)$ pairs, Theorem 4.1 is derived by exploiting the linear structure shared by various $(s, a)$ pairs, creating a novel $\epsilon$-net to

---

**Algorithm 1** DRVI-L

---

1: **Input:** $\underline{\beta}, \mathcal{D} = \{(s_h^\tau, a_h^\tau, r_h^\tau)\}_{\tau,h=1}^{N,H}$.
2: **Init:** $\widehat{V}_H = 0, \widehat{w}_H = 0$.
3: **for** step $h = H$ **to** $1$ **do**
4: $\quad \Lambda_h = \sum_{\tau=1}^N \phi\left(s_h^\tau, a_h^\tau\right) \phi\left(s_h^\tau, a_h^\tau\right)^\top + \lambda I, \quad \widehat{\theta}_h = \Lambda_h^{-1}\left[\sum_{\tau=1}^N \phi(s_h^\tau, a_h^\tau) r_h^\tau\right]$
5: $\quad$ **if** $h < H$ **then**
6: $\quad\quad$ Update $\widehat{w}_{h,i}$ with Equation 6.
7: $\quad$ **end if**
8: $\quad \widehat{\nu}_h = \min(\widehat{\theta}_h + \widehat{w}_h, H - h + 1)_+, \quad \widehat{Q}_h(\cdot,\cdot) = \phi(\cdot,\cdot)^\top \widehat{\nu}_h$
9: $\quad \widehat{\pi}_h(\cdot \mid \cdot) = \arg\max_{\pi_h}\langle \widehat{Q}_h(\cdot,\cdot), \pi_h(\cdot \mid \cdot)\rangle_\mathcal{A}, \quad \widehat{V}_h(\cdot) = \langle \widehat{Q}_h(\cdot,\cdot), \widehat{\pi}_h(\cdot \mid \cdot)\rangle_\mathcal{A}$
10: **end for**

---

control the finite sample error for the entire linear function space, and utilizing the power of our value shift algorithmic ingredient. These techniques are novel compared to the non-robust counterpart.

Specifically, when $\underline{\beta}$ is relatively small, i.e., when the algorithm tends to learn a pessimistic view, we have $\text{SubOpt} = \tilde{O}(d^{1/2}H^{3/2}\underline{\beta}^{1/2}e^{H/\underline{\beta}}/N^{1/2})$. When $\underline{\beta} \to \infty$, i.e., when the algorithm is learning a nearly non-robust view, our bound reduces to $\tilde{O}(d^{1/2}H^2/N^{1/2})$, which recovers the same dependence on $H$ as the non-robust PEVI algorithm in Jin et al. (2021) and achieves the optimal dependence on $N$ and $d$ in Yin et al. (2022). The suboptimality in the $H$ dependency arises as the result of our relatively simple algorithmic design to outline the first step in DRRL with linear function approximation. Adopting the advanced techniques of Yin et al. (2022) could potentially address the discrepancy, and we leave it as a future direction.

## 5 EXTENSIONS

### 5.1 BEYOND UNIFORMLY WELL-EXPLORED DATASET

In practical applications, the data coverage may not satisfy Assumption 4.3, which requires the behavior policy to explore all feature dimensions with a sufficiently high exploration rate. Instead, we only need the behavior policy to adequately cover the features that the optimal policy will visit. To address this, we propose a pessimistic variant of our Algorithm 1, called **P**essimistic **D**istributionally **R**obust **V**alue **I**teration with **L**inear function approximation (PDRVI-L), inspired by the approach in Jin et al. (2021). Under a weaker data coverage condition, sample efficiency can be achieved as long as the dataset sufficiently covers the trajectory induced by the optimal policy $\pi^*$. We formalize this condition in Assumption 5.1.

**Assumption 5.1** (Robust Sufficient Coverage of the Optimal Policy). *Suppose there exists an absolute constant $c^\dagger > 0$ such that for any $P \in \mathcal{P}(\rho)$,*

$$\Lambda_h \geq I + c^\dagger \cdot N \cdot d \cdot \mathbb{E}_{P,\pi^*}[(\phi_i(s_h, a_h)\mathbb{1}_i)(\phi_i(s_h, a_h)\mathbb{1}_i)^\top | s_1 = s],$$

$\forall s \in \mathcal{S}, h \in [H], i \in [d]$, *holds for probability at least $1 - \delta$.*

Compared to the sufficient coverage condition in Jin et al. (2021), our Assumption 5.1 requires the collected samples $\Lambda_h$ to cover each dimensions $i \in [d]$ uniformly well. This requirement arises from the ambiguity set constructed in the latent factor space. Moreover, we require this condition to hold uniformly across all the transition model within the ambiguity set, motivated by Blanchet et al. (2023). We summarize our algorithmic design in Appendix C and present it as Algorithm 2. In contrast to Algorithm 1, we subtract a pessimistic term $\gamma_h \sum_{i=1}^d \|\phi_i(s,a)\mathbb{1}_i\|_{\Lambda_h^{-1}}$ from the estimated $Q$-value in Algorithm 2. This discourages our algorithm from selecting the action with less confidence. Compared to Jin et al. (2021), which uses $\gamma_h\|\phi(s,a)\|_{\Lambda_h^{-1}}$ as the pessimistic term in the non-robust setting, our approach provides a larger penalization and adapts to the distributionally robust nature. Under the partial coverage condition for our dataset, Algorithm 2 achieves sample efficiency, as shown in the following theorem.

**Theorem 5.1.** *In Algorithm 2 we set* $\lambda = 1$ *and*

$$\gamma_h = c_1\underline{\beta}(e^{\frac{H-h}{\underline{\beta}}} - 1)d\zeta_3^{1/2} + c_2\underline{\beta}^{1/2}(e^{\frac{H-h}{\underline{\beta}}} - 1)H^{1/2}\zeta_2^{1/2},$$

*where* $\zeta_2$ *is the same as in Theorem 4.1 and* $\zeta_3 = \log(2N + 32N^2H^3d^{5/2}\zeta e^{2H/\beta})$ *for some absolute constant* $c_1$ *and* $c_2$ *that are only dependent on* $c^\dagger$*. Then under the Assumption 4.1, 4.4 and 5.1, our algorithm 2 has the following guarantee with probability at least* $1 - \delta$*,*

$$\text{SubOpt}(\widehat{\pi}; \mathcal{P}) \leq c_1\underline{\beta}(e^{H/\underline{\beta}} - 1)d^{3/2}H\zeta_3^{1/2}/N^{1/2} + c_2\underline{\beta}^{1/2}(e^{H/\underline{\beta}} - 1)d^{1/2}H^{3/2}\zeta_2^{1/2}/N^{1/2}.$$

Our bound incurs an additional factor of $d$ compared to Theorem 5.2 as a price of the weaker data coverage condition Specifically, the suboptimality for the Algorithm 2 is $\tilde{O}(d^{3/2}H^{3/2}\underline{\beta}^{1/2}e^{H/\underline{\beta}}/N^{1/2})$ when $\underline{\beta}$ is relatively small. When $\underline{\beta} \to \infty$, i.e., the algorithm is learning a nearly non-robust view, the suboptimality reduces to $\tilde{O}(d^{3/2}H^2/N^{1/2})$, which recovers the same dependence on $d$, $H$, and $N$ as Jin et al. (2021). Recently, Yin et al. (2022) improves the suboptimality bound to $\tilde{O}(d^{1/2}H^{3/2}/N^{1/2})$ with a more complex algorithmic design. As our paper is the first attempt to design linear function approximation to solve the offline DRRL problem, we leave the improvement towards the optimal rate as a future direction.

## 5.2 MODEL MISSPECIFICATION

The assumption of state aggregation may not be realistic when applied to real-world datasets. In this subsection, we relax the soft state-aggregation MDP assumption to allow for the possibility of a true transition kernel that is nearly a state-aggregation transition.

**Assumption 5.2** (Model Misspecification in Transition Model)**.** *We assume that for all* $h \in [H]$*, there exists* $\tilde{P}_h \in \text{Span}(\phi)$ *and* $\xi \geq 0$ *such that each* $(s, a)$*, the true transition kernel* $P_h(\cdot|s, a)$ *satisfies* $\|P_h(\cdot|s, a) - \tilde{P}_h(\cdot|s, a)\|_1 \leq \xi$*. For the reward functions, we still assume that* $r_h \in \text{Span}(\phi)$ *for all* $h \in [H]$*.*

**Theorem 5.2** (Model Misspecification)**.** *We set* $\lambda = 1$ *in Algorithm 1. Under the Assumption 4.3, 4.4 and 5.2, when* $N \geq 40/\underline{c} \cdot \log(4dH/\delta)$*, we have the following holds with probability at least* $1 - \delta$*,*

$$\text{SubOpt}(\widehat{\pi}; \mathcal{P}) \leq c_1\underline{\beta}(e^{H/\underline{\beta}} - 1)(\xi d^{1/2} + d^{1/2}\zeta_1^{1/2})H/N^{1/2} + c_2\underline{\beta}^{1/2}(e^{H/\underline{\beta}} - 1)H^{3/2}\zeta_2^{1/2}/N^{1/2} + \xi H^2/2.$$

*Here* $\zeta_1$ *and* $\zeta_2$ *are the same in Theorem 4.1 and* $c_1$ *and* $c_2$ *are some absolute constants that only depend on* $\underline{c}$*.*

**Theorem 5.3** (Model Misspecification with Sufficient Coverage)**.** *In Algorithm 2 we set* $\lambda = 1$ *and*

$$\gamma_h = c_1\underline{\beta}(e^{\frac{H-h}{\underline{\beta}}} - 1)d\zeta_3^{1/2} + c_2\underline{\beta}^{1/2}(e^{\frac{H-h}{\underline{\beta}}} - 1)H^{1/2}\zeta_2^{1/2},$$

*where* $\zeta_2$ *and* $\zeta_3$ *are the same as in Theorem 5.1 and* $c_1, c_2 \geq 1$ *are some absolute constants that only involve* $c^\dagger$*. Then based on Assumptions 4.4, 5.1 and 5.2, our Algorithm 2 has the following guarantee with probability at least* $1 - \delta$*,*

$$\text{SubOpt}(\widehat{\pi}; \mathcal{P}) \leq c_1\underline{\beta}(e^{H/\underline{\beta}} - 1)(\xi d + d^{3/2}\zeta_3^{1/2})H/N^{1/2} + c_2\underline{\beta}^{1/2}(e^{H/\underline{\beta}} - 1)d^{1/2}H^{3/2}\zeta_2^{1/2}/N^{1/2} + \xi H^2/2.$$

According to Theorem 5.2, when the soft-state aggregation model is inaccurate up to $\xi$ total variation, the policy's performance incurs an approximation gap of $O(\xi \cdot (\underline{\beta}(e^{H/\underline{\beta}} - 1)d^{1/2} + H^2))$ and $O(\xi \cdot (\underline{\beta}(e^{H/\underline{\beta}} - 1)d + H^2))$ for our DRVI-L and PDRVI-L algorithms, respectively. The extent of degradation depends on the total-variation divergence of the empirical transition distribution from the true transition distribution, and the desired level of robustness.

## 6 EXPERIMENT

We evaluate the robustness and sample efficiency of our algorithms through numerical experiments in two well-known environments from the robust RL literature: the American put option environment (Tamar et al., 2014; Zhou et al., 2021b) and the CartPole environment in OpenAI Gym (Brockman et al., 2016). The American put option environment showcases the robustness and the impact of

different linear approximators, while the CartPole environment allows us to compare our algorithm with previous methods in a challenging setting with complex dynamics and a higher-dimensional, continuous state space. Additional experimental setup details can be found in Appendix D.

**American Put Option:** We compare Algorithm 2 with its non-robust counterpart, Pessimistic Value Iteration (PEVI) (Jin et al., 2021). Both algorithms are trained in an environment with $p_0 = 0.5$ and evaluated in a perturbed environment with varying $p_0$. The results, shown in Figure 2(a), demonstrate that the robust agent, particularly with a suitable radius $\rho = 0.01$, outperforms the non-robust agent in the perturbed environment with $p > 0.55$, with a slight performance degradation at $p_0 = 0.5$. Next, we investigate the impact of dimension $d$ on suboptimality ($\|\widehat{V}_1 - V_1^*\|$) and computational time. Figure 2(b) reveals that a smaller $d$ leads to lower estimation error and higher approximation error, given the same amount of data. The misspecification of the linear transition model introduces intrinsic bias to value estimation, but an appropriate bias reduces the estimation error with limited data, which is crucial for offline learning. Furthermore, Figure 2(c) demonstrates that the computational cost increases linearly with the dimension, rather than the size of the state-action space, indicating the potential of our algorithm for deployment in large-scale problems.

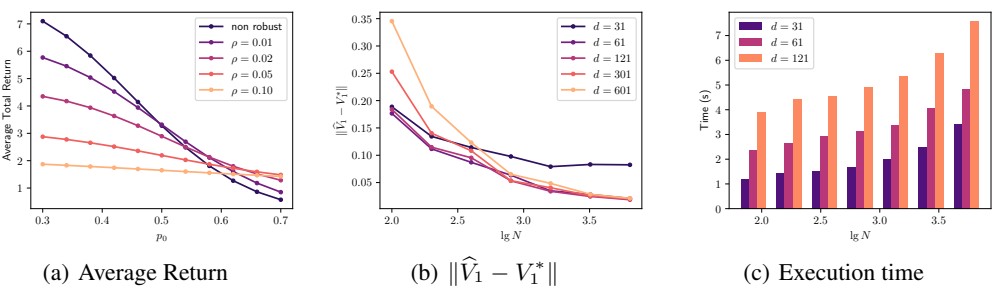

| (a) Average Return | (b) $\|\widehat{V}_1 - V_1^*\|$ | (c) Execution time |

Figure 2: Results in American Option Experiment. (a) Average total return of different KL radius $\rho$ in the perturbed environments ($N = 1000$, $d = 61$). (b) Estimation error with different linear function dimension $d$'s and the sizes of dataset $N$'s ($\rho = 0.01$). (c) Execution time for different $d$'s.

**CartPole:** We compare our PDRVI algorithm with several representative offline RL algorithms in the CartPole environment: (a) RFQI (Panaganti et al., 2022b), known for its capacity for non-linear approximation and superior performance; (b) RAPI (Tamar et al., 2014), further validating our message about the limitations of the RTA approach; (c) PEVI (Jin et al., 2021) as a non-robust benchmark. A summary of their features can be found in Table 3. To evaluate the algorithms' robustness, we introduce different levels of action perturbations and assess their performance in the perturbed environments. Our PDRVI algorithm demonstrate superior robust performance compared to non-robust PEVI and comparable performance to RFQI, despite using a simpler approximator. Notably, our algorithm enjoys theoretical guarantees through the DR variant of pessimism, while RFQI relies on a batch-constrained Q-learning algorithm that may not converge optimally under weak data coverage conditions. In contrast, RAPI performs poorly compared to other algorithms in all cases, supporting our claim in Section 3 that the RTA design can lead to suboptimal decisions. Despite using the more conservative $R$-contamination (R-con) ambiguity set, RAPI's significant performance gap compared to other algorithms confirms the ineffectiveness of the RTA approach.

| **Algo** | **Representation.** | **Pessimism.** | **Ambiguity.** |
|---|---|---|---|
| PDRVI (Ours) | Linear | Yes | KL |
| RFQI | NN | Yes | TV |
| RAPI | Linear | No | R-con |
| PEVI | Linear | Yes | None |

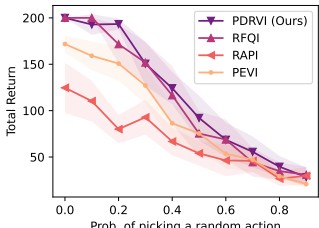

Figure 3: Experiment Results for the CartPole environment. (Left) Summary of the algorithms' features. (Right) Average return of different algorithms in the perturbed environments over 10 random seeds shadowed with standard deviation.

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
