# OpenReview forum: "Robustify the Latent Space: Offline Distributionally Robust Reinforcement Learning with Linear Function Approximation"
_ICLR.cc/2024/Conference — Submitted to ICLR 2024_

### Official Review · Reviewer_p4nt · 2023-10-25

**Soundness:** 2 fair
**Presentation:** 3 good
**Contribution:** 3 good
**Rating:** 5
**Confidence:** 3

**Summary:**

This paper consider offline robust RL for linear MDP with linear function approximation. The work is the first to consider this setting.

**Strengths:**

1. The setting is interesting and novel, especially with function approximation for robust RL.
2. The extension part makes the results more general.

**Weaknesses:**

1. I don't get the method in the work, how do you tackle the uncertainty from the dataset? Like in the tabular setting (Shi & Chi, 2022) and previous offline RL works, a penalty term is subtracted for those less visited states. How do you handle it in your method?
2. The result seems not too surprising to me. The work is under the linear MDP setting, and the uncertainty set is defined w.r.t. $\psi$. This setting is a little bit easy to me, as many nice properties hold under it, like the completeness of the linear function class.

**Questions:**

See the weakness part.

---

> ### Author Response · Authors · 2023-11-22
>
> Thank you for your detailed reading and valuable comments. Please find our response to the comments below:
>
>
> ---
>
> **Q.** The approach to tackle the uncertainty from the dataset.
>
> **A.** We adopt a similar pessimism principle by subtracting $\gamma_h \sum_{i=1}^d \lVert \phi_i(s,a)1_i\rVert_{\Lambda_h^{-1}}$ from the estimated $Q$-value, which is stated in the last paragraph on Page 7. We apologize for any confusion in our presentation and will provide a clearer outline in our next version.
>
> ---
>
> **Q.** The novelty of the results.
>
> **A.** We acknowledge the reviewer's observation that, after understanding the proper construction of the ambiguity set, the subsequent results can generally hold under many favorable properties. The novelty of our work lies in uncovering the proper construction, a previously unknown aspect until recent progress by Goyal, Vineet, and Julien Grand-Clement in 2023. As our paper is the first to establish finite-sample results in this setting, we rely on some strong conditions, like Definition 2.1, to outline the main message. Additionally, we introduce the novel value-shift trick and the corresponding technical skills in proving the theoretical results, which themselves contribute valuable insights.
>
> ---
>
> [1] Goyal, Vineet, and Julien Grand-Clement. "Robust Markov decision processes: Beyond rectangularity." Mathematics of Operations Research 48.1 (2023): 203-226.

---

### Official Review · Reviewer_GkC9 · 2023-10-29

**Soundness:** 2 fair
**Presentation:** 2 fair
**Contribution:** 2 fair
**Rating:** 3
**Confidence:** 4

**Summary:**

This paper proposes to combine the linear MDP with $d$-rectangular uncertainty set to admit linear function approximation for offline distributionally robust RL. Their proposed algorithms incorporate the linear function approximation and have finite-sample suboptimality bounds. The numerical experiment further show the performances.

**Strengths:**

This work proposes a novel setting incorporating linear MDP with $d$-rectangular assumption, in which the state-action value function admits linear representation. This is also the first attempt to investigate finite-sample sample complexities for offline distributionally robust RL with linear function approximation.

**Weaknesses:**

There are severe errors in this paper. The writing is not clear, some parts of the paper are hard to follow. The theoretical results are questionable.

**Questions:**

1. The part around equation (5) is hard to follow. It seems that a simple clip of the estimator from below can prevent the 'approach infinity and damage estimation' problem. I don't see the necessity of using value shift, and don't see how it contributed to the 'improved theoretical results' as claimed in the abstract.
2. For the uniformly well-explored dataset assumption, the authors state that it is adopted in Jin et al. (2021); Duan et al. (2019) and Xie et al. (2021). Can you specify respectively in which assumptions/theorems they use the uniformly well-explored dataset assumption? Moreover, in the proof of Th 5.2, in the end there is a term like this, $\Vert\sum_{i=1}^d\phi_i(s,a)\bf{1}_i\Vert$, which is unique in your setting. But the assumption is in the form of $E[\phi(s_h,a_h)\phi(s_h,a_h)^{\top}]$. The proof is omitted by claiming using similar steps in the proof of Corollar 4.6 in Jin et al. (2021). More details should be provided to show the correctness of this assumption.
3. For assumption 4.4, as far as I know, Zhou et al. (2021b), Panaganti & Kalathil (2022) didn't use this assumption. Actually, assumption 4.4 states that the optimizer $\beta_{h,i}^*$ is lower bounded for any function $V$, this assumption cannot be true even with small uncertainty level $\rho$. Specifically, in proposition 2 of Hu and Hong (2013), we have $\beta^*=0$ if and only if $\kappa:=P(X=essinf X)>0$ and $\log\kappa+\delta\geq 0$. If X is a constant, then $\kappa=1$ and $\log\kappa+\delta\geq 0$ for any $\delta$, in which case $\beta^*$ can be zero. On the other hand, we should choose $\rho$ based on practical need, rather always a small value. Even if it is small, it still cannot guarantee that assumption.
4. Blanchet et al. (2023) also studied the same setting. They proved a $O(d^2H^2/\sqrt{N})$ order suboptimality bound. While in your work,  you proved suboptimality bounds of $d^{1/2}H^2/\sqrt{N}$ and $d^{3/2}H^2/N^{1/2}$. Why your results are $d^{3/2}$ and $d^{1/2}$ better, respectively?
5. On page 8, it seems to me that section 5.2 of model misspecification comes out of no where. There is no model and formulation here, such as what is that uncertainty set for the non-linear transition kernel. In proof, the authors simply bound the inf of two uncertainty sets like as follows:
$$\inf_{P_{h+1} \in \mathcal{P_{h+1}}}E_{P_{h+1}}[V_{h+1}]-\inf_{P_{h+1} \in \tilde{\mathcal{P}}}E_{P_{h+1}}[V_{h+1}] \leq (H-h)\xi.$$ I don't think this is correct. Please provide with more argument.



Typos and lack of clarification:
1. At the beginning of page 2, distributional (DR) robust RL.
2. At the beginning of page 3, the definition of value function and Q-function are wrong. NO definition of $N$, $d$, $\delta$.
3. Misuse of RLS as RTS in page 4.
4. On page 2, how you define 'weak data conditions'. Is there a strong data condition counterpart? In the statement of open problem, what is the 'weak' data coverage conditions, weaker compared to what?
5. Should $\phi(s,a)\geq 0$, which is necessary in the definition of $d$-rectangularity. The current definition in linear MDP cannot ensure that.

---

> ### Author Response · Authors · 2023-11-22
>
> Thank you for your detailed reading and valuable comments. Please find our response to the comments below:
>
> ---
>
> **Q.** The necessity of using value shift to the improved theoretical results.
>
> **A.** The vanilla estimator itself can take any real number. The simple clip can ensure the estimator to be nonnegative, which is still not sufficient to prevent the approach to infinity, as $\log(Z)$ will still go to infinity when $Z$ approaches zero. Thus, the value shift is necessary to ensure the numerical safety of the estimation.
>
> In terms of the theoretical results, without the value shift technique, one can obtain $O(\underline{\beta}e^{H/\underline{\beta}} H^2 / N^{1/2})$ result, and it would approach infinity when $\underline{\beta}$ approaches zero (when the algorithm is learning a nearly non-robust view). In fact, Zhou et al. 2021 establishes the finite-sample guarantee under the KL divergence without using the value shift and ends up with $\underline{\beta}e^{H/\underline{\beta}}$ as a coefficient of their dominating term (See Lemma 5 in Zhou et al. 2021). By introducing the value shift technique, we can obtain $O(\underline{\beta}(e^{H/\underline{\beta}}-1) H^2 / N^{1/2})$ and can succeed in recovering the same dependence on H as the non-robust PEVI algorithm.
>
> ---
>
> **Q.** Uniformly well-explored dataset assumption.
>
> **A.** Corollary 4.6 in Jin et al. 2021, Theorem 2 in Duan et al. 2020, and Theorem 3.2 in Xie, Tengyang, et al. 2021 all depend on $1/\lambda_{\min}(\Sigma_h)$, which, in fact, requires the dataset to be uniformly well-explored.
>
> In terms of the correctness of Assumption 4.3, note that in the proof of Theorem 5.2 on Page 13 in the Appendix,
> $$\lVert \Lambda_h^{-1}\rVert_{\operatorname{tr}(\phi(s,a))}\le \lVert \Lambda_h^{-1}\rVert^{1/2} \sum_{i=1}^d \lVert \phi_i(s,a) 1_i\rVert.$$ In fact, $\sum_{i=1}^d \lVert \phi_i(s,a) 1_i\rVert = 1$ due to our Definition 2.1, and thus the key to this proof is to upper bound the $\lVert \Lambda_h^{-1}\rVert^{1/2}$ with sufficient high probability.
> Note that $\Lambda_h$ is N samples with mean as $E[\phi(s_h, a_h)\phi(s_h, a_h)^\top]$ and thus we can apply concentration argument to upper bound it, which is the same as the proof of Corollary 4.6 in Appendix B.4 in Jin et al. 2021. We will add the full proof to show the correctness of this assumption in the next version.
>
> ---
>
> **Q.** The justification of Assumption 4.4.
>
> **A.** Assumption 4.4 is common in literature when considering KL divergence. In fact, Zhou et al. (2021b) heavily relies on Assumption 4.4, as their results involve $e^{R_\max/\underline{\beta}}$ in their constant $C_{\pi}$ and $C$ (See their proof of Lemma 5, Theorem 1, and Theorem 2 in the supplement material). Panaganti & Kalathil (2022) only consider the total-variation distance instead of KL divergence. We agree with the reviewer's understanding of the relationship between $\rho$ and $\beta^*$. By the completion of this paper, all existing literature relies on the same or similar assumption for finite-sample results under KL divergence. Yang, Wenhao et al. 2021 does not rely on this assumption. Instead, their results depend on $1/\underline{p}$ where $\underline{p} = \min_{p(s'\lvert s,a)>0} p(s'\lvert s,a)$ and is also restricted in some cases.
>
> ---
>
> **Q.** Correctness of the dependence on the dimension.
>
> **A.** The dependence on the dimension should be $d^{2}$ for the Sufficient Coverage case, which matches Blanchet et al. (2023). This error comes from the fifth inequality on Page 19, where $\sum_{i=1}^d \sqrt{\frac{1}{1+c^{\dagger}N}}$ should be upper bounded by $d$, rather than $d^{1/2}$. However, for the well-explored setting, the dependence on $d$ is indeed $d^{1/2}$, as the condition is stronger than sufficient-coverage, and by Definition 2.1, the dependence reduces to bounding $\sum_{i=1}^d \lVert \phi_i(s,a)1_i\rVert =1$.
>
> ---
>
> **Q.** Correctness of the dependence on the model misspecification.
>
> **A.** We agree with the reviewer's understanding. The transformation from the error in the transition model $\xi$ to the final suboptimality is not a one-step proof. In fact, we need to follow some parts of the proof when non-model misspecification happens, and only finite-sample error occupies. We will update the results accordingly.
>
> ---
>
> **Q.** Typos.
>
> **A.** We will correct the definition of the value function and the $Q$ function as $V_h^{\pi}(s,a) = E_{P}^{\pi}[\sum_{h'=h}^H r_{h'}(s_{h'}, a_{h'})\lvert s_{h} = s]$ and $Q_h^{\pi}(s,a) = E_{P}^{\pi}[\sum_{h'=h}^H r_{h'}(s_{h'}, a_{h'})\lvert s_{h} = s, a_{h} = a]$, as well as the misuse of RLS as RTS.
>
> ---
>
> **Q.** Stronger data condition vs. weak data condition.
>
> **A.** A strong data condition refers to our Assumption 4.3 (Uniformly Well-explored Dataset). Weak data condition refers to Assumption 5.1 (Sufficient Coverage).

---

> > ### Author Response · Authors · 2023-11-22
> >
> > **Q.** The necessity of the non-negativity of the feature map $\phi(s,a)$.
> >
> > **A.** Yes, we should impose $\phi(s,a)\ge 0$ in Definition 2.1, and we will add it in the next version.
> >
> > ---
> >
> > [1] Ying Jin, Zhuoran Yang, and Zhaoran Wang. Is pessimism provably efficient for offline rl? In
> > International Conference on Machine Learning, pp. 5084–5096. PMLR, 2021.
> >
> > [2] Duan, Yaqi, Zeyu Jia, and Mengdi Wang. "Minimax-optimal off-policy evaluation with linear function approximation." International Conference on Machine Learning. PMLR, 2020.
> >
> > [3] Xie, Tengyang, et al. "Bellman-consistent pessimism for offline reinforcement learning." Advances in neural information processing systems 34 (2021): 6683-6694.
> >
> > [4] Panaganti, Kishan, et al. "Robust reinforcement learning using offline data." Advances in neural information processing systems 35 (2022): 32211-32224.
> >
> > [5] Yang, Wenhao, Liangyu Zhang, and Zhihua Zhang. "Toward theoretical understandings of robust Markov decision processes: Sample complexity and asymptotics." The Annals of Statistics 50.6 (2022): 3223-3248.

---

### Official Review · Reviewer_WFAe · 2023-10-30

**Soundness:** 3 good
**Presentation:** 2 fair
**Contribution:** 3 good
**Rating:** 6
**Confidence:** 4

**Summary:**

This paper studies distributionally robust offline linear soft state aggregation MDP under d-rectangularity condition with a KL distance to measure the probability distance. It gives an example why the $(s,a)$-rectangularity condition is not suitable for such problem and well motivates the use of d-rectangularity set. The paper proposed a value iteration method to solve the problem and demonstrate sub-optimality gap under the well-explored regime, the partly explored regime, and the misspecification regime. Numerical results demonstrates the effectiveness of the proposed method.

**Strengths:**

The paper provides a full story from the motivation, problem setting, algorithm design, complexity analysis, and several extensions to cover different regimes in practices. Certain tricks like value shift technique are used to bypass challenges in estimations, which can be of independent interests.

**Weaknesses:**

This is overall a strong paper.

In terms of presentation, the notation system is very complicated that one can easily get lost, which is also common for distributionally robust RL paper.

In terms of novelty, the motivation and settings are new. However, the components of algorithm design seem to be standard except for the estimation trick. Not a big matter given the more realistic setting.

In terms of result, Yin et al 2022 has improved complexity bound in the non-robust setting, which has been available for more than a year. According to the conjecture of the authors, they should be able to achieve optimal dependency in H as well. The authors are welcomed to add this to the camera-ready version.

**Questions:**

1. I suppose $\psi_{h,i}(s)$ represents the $i$-th coordinate of $\psi_{h}(s)$? Can the author explain why $\psi$ satisfies the second equation in Definition 2.1?

2. Page 2 line 2, it should be "offline distributionally robust (DR) RL".

3. Robustness by definition already means being pessimistic. I wonder what is the different of pessimistic and robustness in the algorithm PDRVI-L?

4. Does the technique and result extend to infinite horizon discounted MDP and why?

---

> ### Author Response · Authors · 2023-11-22
>
> Thank you for your detailed reading and valuable comments. Please find our response to the comments below:
>
> ---
>
> **Q.** Explain $\psi$ in Definition 2.1.
>
> **A.** $\phi_{h,i}(s)$ represents the $i$-th coordinate of $\psi_h(s)$. To understand the second equation in Definition 2.1, it can also be understood as the i-th hidden state among the total d hidden states. $\psi_{h,i}(s)$ can be understood as the probability to transition from the i-th hidden state to the original state $s$.
>
> ---
>
> **Q.** What is the difference between pessimistic and robustness in the algorithm PDRVI-L.
>
> **A.** Pessimism is the general learning principle for designing sample-efficient algorithms that can overcome poor data coverage, i.e., the finite samples in the offline dataset are not diverse enough to cover sufficient state-action pairs. This is achieved by subtracting $\gamma_h \sum_{i=1}^d \lVert \phi_i(s,a) 1_i\rVert_{\Lambda_h^{-1}}$ from the estimated $Q$-value. Robustness is aimed at tackling the possible distribution mismatch from the training dataset to the test environment.
>
> ---
>
> **Q.** Extension to the infinite horizon discounted MDP.
>
> **A.** Most of the techniques, including the algorithmic design as well as the finite-sample analysis of the KL divergence, can be extended to the infinite horizon discounted MDP. The gap comes from transferring the finite-sample error from the estimation of the KL ambiguity set to the policy value. While in the finite horizon settings, each stage is independent and can be analyzed backward, in the infinite discounted MDP, we need to analyze the sensitivity of the fix-point solution of the Bellman optimality equation to the finite-sample error .

---

> > ### Comment · Reviewer_WFAe · 2023-12-04
> >
> > I would like to thank the authors' response and encourage the authors to improve notation system and the presentation.

---

### Official Review · Reviewer_T9sS · 2023-10-30

**Soundness:** 2 fair
**Presentation:** 1 poor
**Contribution:** 2 fair
**Rating:** 3
**Confidence:** 4

**Summary:**

This work targets the problem of handling model distribution shifts in offline RL problems, especially in the case that the environment is modeled in a linear function approximation way. This is an interesting problem since most of the existing works for this problem are in tabular settings. Targeting two history dataset quality: well-explored or weaker coverage, it proposed a robust algorithm DRVI-L, as long as the non-asymptotic sample complexity guarantees for it. In addition, this work conducted experiments to evaluate the performance of DRVI-L with comparisons to existing robust RL algorithms and non-robust counterparts.

**Strengths:**

1. The topic of offline robust RL with linear function approximation is very interesting and in a pressing need.
2. It provides a framework to target this problem by introducing proper assumption (Soft State Aggregation) and designing new algorithms based on this setting.
3. The algorithm has been tested in experiments and shows competitive results compared to existing algorithms.

**Weaknesses:**

1. There exist many confusing notations or not proper abuse of notations. For example, $R^{rob} (\pi, P) $ is abbreviated as $R(\pi, P)$, which actually already represents the non-robust expected return, which causes some confusion about the notation $R$ later. More are listed later in details.
2. The technical correctness is not clear to the reviewer since the Lemma E.2 in Appendix seems not correct. The reason is that: firstly, in Lemma E.1, there exists some transition kernel $P\_{exist} \in \mathcal{P}\_1$ (denoted as $P_{1,h}^*$ in the paper) that can leads to the equality in (14). However, this applying $P_{1,h}^*$ also defined in a  $\arg\min$ notion below (13). I guess actually $P_{1,h}^*$ is just some transition kernel in $\mathcal{P}_{1}$ but not in a $\arg\min$ notion. Then applying Lemma E.1 to Lemma E.2 won't leads to the results that this work claims. Since Lemma E.1 and E.2 are essential part in the proof, it makes the technical correctness doubtful for the reviewer.

**Questions:**

1. As mentioned, what does the notation $P_{1,h}^*$ and $P_{2,h}^*$ mean in the proof of Lemma E.1, it has been re-defined at least twice.
2. The expectation over $P_{1,\pi}^*$ in Lemma E.1 is not clear. At each time step $h$, the expectation is determined by  $P_{1,h,\pi}^*$ according to the reviewer's understanding. Carefully polishing the notation and presentation will be very helpful for the reader to check and follow the proof.
2. What does $P^*$ mean in the proof of Lemma E.2, which has not been mentioned?

---

> ### Author Response · Authors · 2023-11-22
>
> Thank you for your detailed reading and valuable comments. Please find our response to the comments below:
>
> ---
>
> **Q.** Confusion about the notations
>
> **A.** We sincerely apologize for the confusion caused by the excessive use of notations. We will clarify them one by one in the following:
>
> 1. $R^{rob}(\pi, P)$ is abbreviated and used as $R(\pi, P)$ since we no longer discuss the non-robust return and only focus on the distributionally robust one.
>
> 2. We acknowledge the reviewer's understanding of $P_{1,h}^*$. To be precise, $P_1^*$ is a transition kernel in $\mathcal{P}_{1,h}(s,a)$ satisfying the last equality of Equation (14). The two arg min notions below (13) and the $P_2^*$ notion are supposed to be removed. In fact, Lemma E.2 should be corrected, as $P_h^*$ and $\widehat{P}_h^*$ are transition kernels in $\mathcal{P}_h(s,a)$ and $\widehat{\mathcal{P}}_h(s,a).$ Sorry for the typo from the older version, but the proof is established for the correct statement.
>
> 3. We agree with the reviewer's understanding of the expectation over $\mathcal{P}_{1,\pi}^{*}$ and will polish the notation in the next version.
>
> 4. $P^*$ denotes a transition kernel in $\mathcal{P}$.

---

> > ### Comment · Reviewer_T9sS · 2023-12-04
> > **Response to the author**
> >
> > Thanks for the author's response. Although the author explains that the concern of the correctness of the proof induced by the confusion about the notation, a fully updated and polished proof will be helpful for the readability of this work, also allowing all the reviewers to check the correctness. So the reviewer still believes there is some room to improve the presentation till convincing. Thanks!

---

### Official Review · Reviewer_EUgk · 2023-11-01

**Soundness:** 3 good
**Presentation:** 3 good
**Contribution:** 2 fair
**Rating:** 5
**Confidence:** 4

**Summary:**

This paper considers offline RL in a training-test model mismatch. They consider linear MDP model and provide robustness guarantees using distributional robust optimization under two settings of offline data: well=explored data and weakly-explored data. They provide sample complexity for robust offline RL for each of the settings of the offline data coverage.

**Strengths:**

* The sample complexity looks reasonable.

**Weaknesses:**

The problem setting is limited
* What if $D$ is not KL? What motivates $D$ to be KL in practice?
* How do we know $\rho$ in practice? What if we misspecified $\rho$?
* In what problems when this kind of model mismatch actually happen? Is there any motivating real example?

**Questions:**

Please see my questions above.

---

> ### Author Response · Authors · 2023-11-22
>
> Thank you for your review and positive comments on our paper. Please find our response to the comments below:
>
> ---
>
> **Q.** What if it is not KL? What motivates it to be KL in practice?
>
> **A.** Our algorithm, as well as the theoretical results, can easily be extended to other divergences, such as $\chi^2$ divergence and total variation distance, by simply replacing the dual function of KL distance in Equation (3) with those of other distances. The analysis of other $f$-divergences is easier than that of the KL divergence, as KL involves high nonlinearity concerning expectation and exhibits no Lipschitz when the dual variable gets close to zero. A similar example to generalize the results under KL divergence to other distances can be found in Yang, Wenhao et al. 2022. We present the results under KL divergence to facilitate the presentation and keep the reader's focus on our main contributions, primarily in our algorithmic design and various data coverage extensions.
>
> ----
>
> **Q.** How do we know $\rho$ in practice? What if we $\rho$ misspecified?
>
> **A.** Finding the optimal ambiguity set radius in distributionally robust optimization (DRO) involves a balance between robustness and conservatism. There isn't a one-size-fits-all approach, as the choice of the ambiguity set radius depends on the specific problem and the available information about the uncertainty in the data. Statistical methods such as cross-validation or bootstrapping can be employed to estimate the uncertainty in your data, as seen in Wang, Jie et al. 2021. A more advanced approach is to adopt Bayesian inference, and in general, tuning $\rho$ itself is an interesting research direction that we leave for further work. If $\rho$ is misspecified, the algorithm may fail to achieve sufficient robustness or may tend to be overly conservative.
>
> ----
>
> **Q.** In what problems does this kind of model mismatch actually happen? Is there any motivating real example?
>
> **A.** Any domain where the dynamics of the data could change gradually or even rapidly would be an ideal testbed for our algorithm. Even when the dynamics remain consistent between the training environment and the test one, there may be insufficient data for accurate enough estimation of the dynamics, leading to non-negligible finite-sample error. For example, in power control schemes, the algorithm is optimized for estimated parameters, such as wireless channel quality (Zhang, Haijun, et al. 2020); or in financial portfolio management, financial data always exhibits high non-stationarity (Huang, Biwei, et al. 2020).
>
> ---
>
> [1] Yang, Wenhao, Liangyu Zhang, and Zhihua Zhang. "Toward theoretical understandings of robust Markov decision processes: Sample complexity and asymptotics." The Annals of Statistics 50.6 (2022): 3223-3248.
>
> [2] Wang, Jie, Rui Gao, and Yao Xie. "Sinkhorn distributionally robust optimization." arXiv preprint arXiv:2109.11926 (2021).
>
> [3] Gupta, Vishal. "Near-optimal Bayesian ambiguity sets for distributionally robust optimization." Management Science 65.9 (2019): 4242-4260.
>
> [4] Zhang, Haijun, et al. "Power control based on deep reinforcement learning for spectrum sharing." IEEE Transactions on Wireless Communications 19.6 (2020): 4209-4219.
>
> [5] Huang, Biwei, et al. "Causal discovery from heterogeneous/nonstationary data." The Journal of Machine Learning Research 21.1 (2020): 3482-3534.

---

> ### Comment · Reviewer_EUgk · 2023-12-04
>
> Thank you for the rebuttal. The assumption of the knowledge of $\rho$ is the main issue I’ve generally had with DRO. This parameter is sensitive and misspecifying this parameter basically makes the framework not useful anymore. I encourage the authors to discuss and emphasize this issue clearly in the manuscript.
>
> My third question was about the model mismatch of $\psi$ in the linear mdp. Why does distribution shift only happen with $\psi$ but not $\phi$? This assumption looks quite unnatural to me, more as a way to sidestep the results of Tamar et al 14 and to fit it the linear mdp, rather than motivated by real problems about model mismatch.

---

### Meta-Review · Area_Chair_KU6G · 2023-12-05

**Metareview:**

This paper explores distributionally robust MDP with linear function approximation, assuming that the reward and true MDP are linear in terms of the feature vector. The authors construct the uncertainty set in the latent space of the feature vectors, resulting in the d-rectangular uncertainty set. They propose a LSVI-type algorithm for solving the Q function and conduct theoretical analyses under conditions such as the existence of a uniformly well-explored dataset, sufficient coverage of the optimal policy, and model misspecification. While the addressed problem is interesting and the investigation of distributionally robust RL with linear function approximation is underexplored, the current submission exhibits immaturity in several aspects. Multiple reviewers point out fatal errors in various parts of the proof, casting doubt on the results and claims. Additionally, some proofs lack clarity and require revision. The authors are strongly encouraged to refine their writing and proofs in response to the discussions and reviews. They should ensure the correctness of the proofs and address the identified issues comprehensively.

**Justification For Why Not Higher Score:**

This paper faces issues in the proof and analysis.

**Justification For Why Not Lower Score:**

N/A

---

### Decision · Program_Chairs · 2024-01-16

Reject